# A shift in anterior–posterior positional information underlies the fin-to-limb evolution

**Koh Onimaru[1,2], Shigehiro Kuraku[3], Wataru Takagi[4], Susumu Hyodo[4], James Sharpe[2,5]\*, Mikiko Tanaka[1]\***

[1]Graduate School of Bioscience and Biotechnology, Tokyo Institute of Technology, Yokohama, Japan; [2]EMBL-CRG Systems Biology Research Unit, Centre for Genomic Regulation, and Universitat Pompeu Fabra, Barcelona, Spain; [3]Phyloinformatics Unit, RIKEN Center for Life Science Technologies, Kobe, Japan; [4]Laboratory of Physiology, Atmosphere and Ocean Research Institute, The University of Tokyo, Chiba, Japan; [5]Institució Catalana de Recerca i Estudis Avançats, Barcelona, Spain

**Abstract** The pectoral fins of ancestral fishes had multiple proximal elements connected to their pectoral girdles. During the fin-to-limb transition, anterior proximal elements were lost and only the most posterior one remained as the humerus. Thus, we hypothesised that an evolutionary alteration occurred in the anterior–posterior (AP) patterning system of limb buds. In this study, we examined the pectoral fin development of catshark (*Scyliorhinus canicula*) and revealed that the AP positional values in fin buds are shifted more posteriorly than mouse limb buds. Furthermore, examination of *Gli3* function and regulation shows that catshark fins lack a specific AP patterning mechanism, which restricts its expression to an anterior domain in tetrapods. Finally, experimental perturbation of AP patterning in catshark fin buds results in an expansion of posterior values and loss of anterior skeletal elements. Together, these results suggest that a key genetic event of the fin-to-limb transformation was alteration of the AP patterning network.

\*For correspondence: james. sharpe@crg.eu (JS); mitanaka@ bio.titech.ac.jp (MT)

**Competing interests:** The authors declare that no competing interests exist.

## Introduction

Regulatory interactions between transcriptional factors play important roles for interpreting a morphogen gradient as positional information (*Balaskas et al., 2012*). Changes in these regulatory interactions may therefore be key players for patterning changes during morphological evolution. The fin-to-limb transformation is a prominent but still unsolved example of morphological evolution. 150 years ago Carl Gegenbaur subdivided the skeletal elements of shark pectoral fins into three segments along the anterior–posterior (AP) axis: propterygium, mesopterygium, and metapterygium (*Gegenbaur, 1865*) (*Figure 1A*), which are also found in the majority of chondrichthyans, none-teleost actinopterygians, placoderms, and acanthodians (*Orvig, 1962*; *Coates, 1994*, *2003*). Therefore, possession of propterygium, mesopterygium, and metapterygium is considered to be a plesiomorphic state for gnathostomes. In the sarcopterygians (lobe-finned fishes including tetrapods), the propterygium and mesopterygium have been lost (*Coates, 2003*), thus, suggesting that anterior positional values have been lost or reduced during tetrapod evolution.

In mouse limb buds, *Hand2*, *Gli3*, and *Shh* are key genes for controlling AP patterning (*Riddle et al., 1993*; *Te Welscher et al., 2002*). One of the earliest patterning events is the mutual

**eLife digest** Humans, mice, and other animals with four limbs belong to a group of land-dwelling animals known as the tetrapods. This group of animals evolved from ancient fish and one crucial adaptation to life on land involved the modification of fins to form limbs. The front pair of limbs (the 'arms') evolved from the 'pectoral' fins of the ancient fish. These fins contain numerous bones that fan out from a set of bones called the pectoral girdle. However, most of the bones nearer the front side (the thumb side in the human limb) were lost in the ancestors of tetrapods as they moved onto land. Only the bone nearest the back remained as the 'humerus', which forms the upper part of the limb (i.e., the upper arm of humans).

In the embryos of mice and other animals, the limbs develop from structures called limb buds. For the limb to develop properly, the cells in the limb bud need to receive specific instructions that depend on their position in the bud. A protein called Gli3R provides cells with information about their position along the 'anterior–posterior' (or thumb-to-little finger) axis of the bud. This protein regulates several genes that are involved in limb development, and this results in different genes being expressed in cells along the anterior–posterior axis. For example, *Alx4* is only expressed in a small area at the anterior end of the bud, while *Hand2* expression is found in a large area towards the posterior part.

Gli3R is also found in a fish called the catshark, but it is not clear how it controls the formation of fins. Onimaru et al. show that the pattern of gene expression in the catshark fin bud is different to that of the mouse limb bud. For example, *Alx4* is expressed in a larger area of the fin bud that extends further towards the posterior, while *Hand2* is only found in a much smaller area at the posterior end of the bud. The experiments also suggest that Gli3R is active in a much larger area of the fin bud than in the limb bud.

Next, Onimaru et al. used a drug on the catshark embryos to increase the activity of another protein that can inhibit Gli3R. The fin buds of these shark had anterior shift in several gene expression domains, and the fins that formed were missing several anterior bones and had only a single bone connected to the pectoral girdle. Onimaru et al.'s findings suggest that during the evolution of the tetrapods, there may have been a shift in the anterior–posterior patterning of the fin bud to form a limb. An important area for future work will be to use genome-wide studies to study the fin/limb buds of other species.

transcriptional repression between *Gli3* in the anterior tissue and *Hand2* in the posterior (*Te Welscher et al., 2002*; *Osterwalder et al., 2014*). This early polarity in expression contributes to the subsequent posterior localized expression of *Shh*, and this morphogen in turn reinforces the anteriorly restricted *Gli3* protein activity (Shh inhibits the default processing of the Gli3 protein to its repressor form (Gli3R), thus, creating a gradient of Gli3R along the AP axis) (*Wang et al., 2000*). Several studies on fin development of actinopterygians and chondrichthyans have revealed that posterior *Shh* expression is conserved among gnathostomes (*Dahn et al., 2007*; *Davis et al., 2007*; *Yonei-Tamura et al., 2008*; *Sakamoto et al., 2009a*). However, in fish fin development, the detailed roles of Shh signalling for AP patterning are not well studied and the role of the Hand2-Gli3 mutual interaction remains to be elucidated.

## Results and discussion

To investigate changes in AP patterning during the fin-to-limb transition, we first cloned a number of AP patterning genes from the non-model species *Scyliorhinus canicula* (*Figure 1B–H* and *Figure 1—figure supplement 1* for phylogentic analyses). In the mouse limb bud, *Alx4*, *Pax9*, *Hand1*, and *Zic3* are positively regulated by Gli3R (*Te Welscher et al., 2002*; *Fernandez-Teran et al., 2003*; *McGlinn et al., 2005*; *Vokes et al., 2008*) and thus are expressed in a localized anterior domain (one-third of the axis), while *Hand2* and *Tbx2* show broad posterior expression domains (two-thirds and one-half of the axis, respectively). In stage 30 *S. canicula* embryos (staged according to *Ballard et al., 1993*), we found instead that the anterior genes *Alx4*, *Pax9*, *Hand1*, and *Zic3* were expressed in broad domains, which extend more posteriorly than in the mouse (half the fin bud for *Alx4*, two-thirds for *Pax9* and *Hand1*, and the whole axis for *Zic3*, *Figure 1B–E*). By contrast, the *Hand2* and *Tbx2*

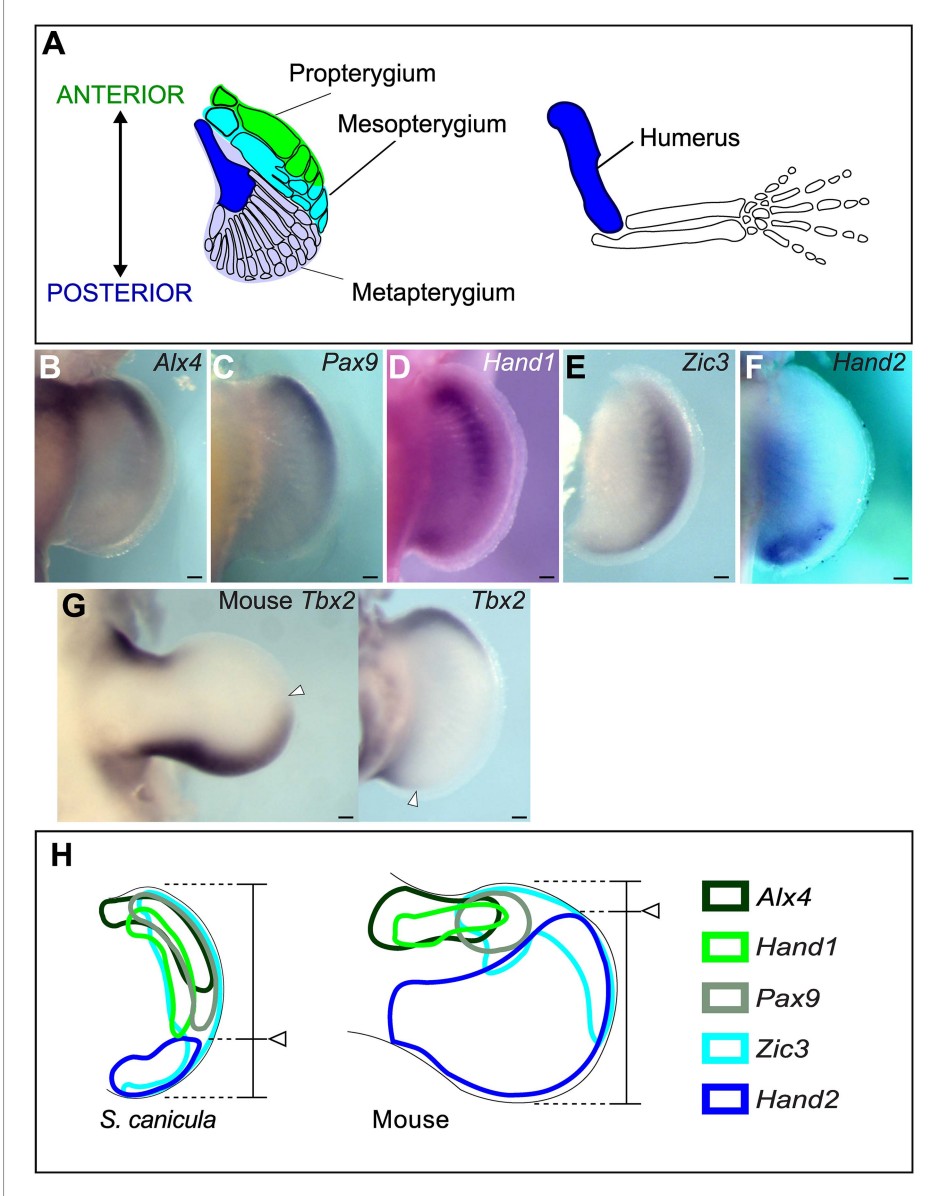

**Figure 1**. Anterior–posterior patterning in *Scyliorhinus canicula* pectoral fin buds. (**A**) Skeletal patterns of *S. canicula* pectoral fin and mouse limb. Blue colours, homologous elements. (**B–G**) In situ hybridisation for *Alx4* (**B**), *Pax9* (**C**), *Hand1* (**D**), *Zic3* (**E**), *Hand2* (**F**), and *Tbx2* (**G**) in *S. canicula* pectoral fin buds at stage 30 and mouse limb bud at E11.5 (left panel in **G**). Arrowheads in **G**, anterior boundary of posterior *Tbx2* expression. Dorsal view; anterior is to the top. Scale bars, 100 μm. (**H**) Schematic of the gene expression patterns. Arrowheads, *Hand2* expression boundary. Expressions of mouse limb buds at E11.5 are after EMBRYS database (*Yokoyama et al., 2009*; *Fernandez-Teran et al., 2003*).

The following figure supplements are available for figure 1:

**Figure supplement 1**. Molecular phylogenetic trees of relevant *S. canicula* genes.

**Figure supplement 2**. Temporal expression analysis of *Alx4*, *Pax9*, *Hand2*, and *Hoxa13* in *S. canicula* pectoral fins.

**Figure supplement 3**. Temporal expression analysis of *Alx4*, *Pax9*, *Hand2*, and *Hoxa13* in chick limb buds.

domains were more posteriorly restricted in *S. canicula* fin buds than in mouse limb buds (*Figure 1F,G*). All of these 6 AP patterning genes show the same trend—their expression boundaries are more posterior in *S. canicula* fin buds (*Figure 1H*), apparently reflecting a gross shift in the AP coordinate system. We chose 3 of these genes to test at multiple time-points to determine whether this was a transient gene expression state (*Figure 1—figure supplements 2, 3*), but in all cases these shifts were observed from stage 29 to stage 31 (which covers ~30 days of *S. canicula* development). In particular, stage 29 is a stage where *Sox9* expression (a prechondrogenic marker) starts in the proximal part of the pectoral fin buds (*Figure 1—figure supplement 2E*), which suggest that the observed shift of AP values would affect proximal skeletal elements as well as distal.

Since the above AP patterning genes are regulated by Shh–Gli3 pathway (*Te Welscher et al., 2002*; *Fernandez-Teran et al., 2003*; *McGlinn et al., 2005*; *Galli et al., 2010*), we cloned *Gli3* from *S. canicula* fin buds (*Figure 2A* and *Figure 2—figure supplement 1A* for phylogenetic tree) and analysed its expression in pectoral fin. In striking contrast to tetrapod limb buds (*Büscher et al., 1997*; *Schweitzer et al., 2000*), *Gli3* expression is not restricted to the anterior region—thus again indicating a general posterior shift of AP positional values in the *S. canicula*. To address whether this situation is conserved in other chondrichthyans, we also cloned and analysed the expression of *Gli3* in pectoral fin buds of a holocephalian, *Callorhinchus milii*, which has propterygium (*Figure 2B*), and again found expression in the posterior part of pectoral fin bud at stage 31 (staged according to *Didier et al., 1998*; *Figure 2B*). Since *Hand2* is expressed posteriorly and thus now overlaps with *Gli3*, this strongly suggests that the Hand2–Gli3 mutual inhibition seen in tetrapods is weak or non-existent in chondrichthyans.

In chick and mouse limb buds, *Gli2* does not play a major role in AP patterning because of its weak processing efficiency to produce its repressor form (*Wang et al., 2000*). However, in zebrafish, *Gli2* does indeed act as a repressor (*Maurya et al., 2013*), so we checked whether *Gli2* could be playing the repressor role in *S. canicula* fin buds. First, we analysed *Gli2* expression in *S. canicula* embryos and found it to be uniform until stage 29 (*Figure 2C*) and then subsequently restricted to the posterior region (*Figure 2C*). Second, we checked whether Gli3 and Gli2 of *S. canicula* have the repressor function, by measuring their processing efficiencies. We analysed the processing determinant domain (PDD), which determines the differential processing efficiencies of Gli3 and Gli2 in mice and humans (*Pan and Wang, 2007*). We inserted the PDDs from human *Gli2* and *S. canicula Gli2* or *Gli3* into the human *Gli3* PDD region (*Figure 2D* and *Figure 2—figure supplement 1* for the amino acid sequences), transfected these constructs into HEK293 cells, and treated the cells with forskolin (FSK) to induce Gli processing. Human and *S. canicula* Gli2 PDD did not induce Gli3R, whereas their Gli3 PDDs did (*Figure 2E*). Thus, in *S. canicula* (as in chick and mouse), Gli3, but not Gli2, plays the major role in repressor production.

We next wished to explore if a genetic explanation could be found for the lack of *Gli3* repression in the posterior part of pectoral fin buds of *S. canicula* and *C. milii*. To compare *Gli3* enhancers in chondrichthyans and tetrapods, we used the VISTA enhancer browser (*Visel et al., 2007*) and found a limb-specific *Gli3* enhancer, element 1586, which replicates anterior *Gli3* expression in mouse limb buds. We identified the homologues of element 1586 in *S. canicula* and *C. milii* and compared them with those from other vertebrates. Consistent with the slow evolutionary rate of chondrichthyans and coelacanth (*Amemiya et al., 2013*; *Renz et al., 2013*; *Venkatesh et al., 2014*), element 1586 is conserved in tetrapods, coelacanth, and chondrichthyans, but not in gar, medaka, and zebrafish (*Figure 3A*). To assess whether the element 1586 in different species has different functionalities, we cloned this element from chick, *S. canicula*, and *C. milii* in front of a basal promoter followed by a *GFP* reporter (*Ochi et al., 2012*; *Figure 3B*). These constructs were electroporated into chick forelimb buds with a constitutively active *RFP* vector (to determine the spatial efficiency of electroporation). As with endogenous *Gli3* expression (*Büscher et al., 1997*), the chick element 1586 drove *GFP* expression specifically in anterior tissue and was repressed in the posterior region, even though *RFP* was expressed throughout the buds (*Figure 3C*). The element 1586 from both *S. canicula* and *C. milii* also drove *GFP* expression in the chick limb buds, confirming that its general activity is conserved from sharks to tetrapods. However, in both cases, the specific posterior repression observed in the chick element was absent (*Figure 3D,E*). Thus, the differential activity of this enhancer (with tetrapods showing posterior repression, and chondrichthyans not) recapitulates the differences in *Gli3* expression within these groups. Furthermore, by recombining *S. canicula* and chick enhancers, we identified a sequence that can exert the posterior repression when inserted into the *S. canicula* enhancer (*Figure 3F,G* and *Figure 3—figure*

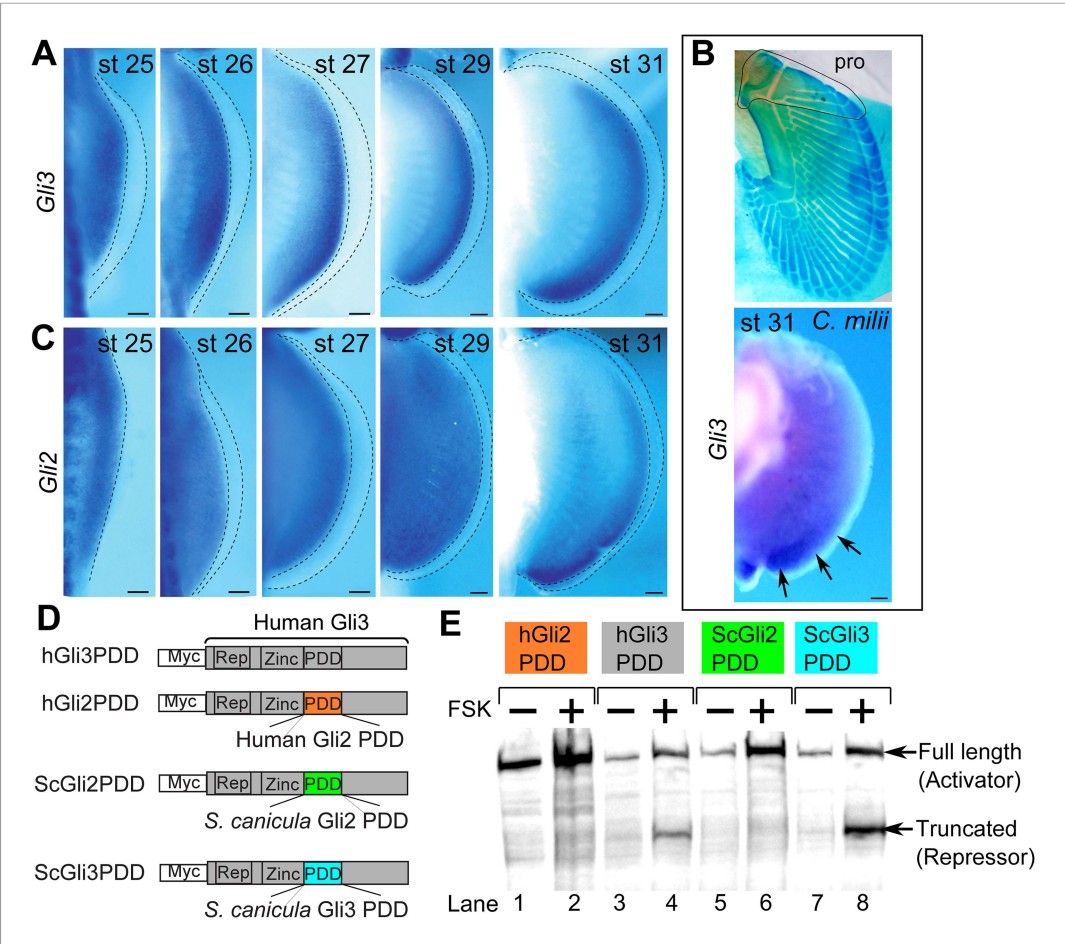

**Figure 2**. Expression and processing of *Gli3* and *Gli2* in *S. canicula* embryos. (**A**) Expression of *Gli3* in *S. canicula* pectoral fins. (**B**) Alcian blue staining of *C. milii* pectoral fin at stage 35 (top, the ventral view of a right fin flipped horizontally) and *Gli3* expression at stage 31 (bottom, a left pectoral fin flipped horizontally). pro, propterygium. (**C**) Expression of *Gli2* in *S. canicula* pectoral fin buds. Scale bars, 100 μm. (**D**) The *Gli3* chimera constructs. hGli3 PDD, full-length human *Gli3* (grey box) with Myc tags. hGli2, ScGli2 and ScGli3 PDD, chimeric *Gli3* genes recombined at the processing determinant domain (PDD) with human *Gli2*, *S. canicula Gli2* and *Gli3*, respectively. (**E**) Protein processing of the chimeric constructs in cell cultures treated with either FSK (+) or DMSO (−). Truncated Gli3 is detected only in hGli3 PDD (lane 4) and ScGli3 PDD (lane 8).

The following figure supplement is available for figure 2:

**Figure supplement 1**. Phylogenetic tree of Gli2 and Gli3, and PDD amino acid sequences.

*supplement 1*). This sequence contains tetrapod or sarcopterygian-specific sequences, suggesting that the posterior repressive activity would have been acquired in a stepwise fashion.

Finally, we wished to address whether changes to AP positional information could modify skeletal arrangement of the propterygium and mesopterygium in catshark. For this purpose, we explored methodologies for performing manipulative experiments on this very slow-developing non-model fish (see 'Materials and methods'). We treated *S. canicula* embryos with retinoic acid (RA) to increase Shh-signalling activity (at stage 29 with 1 μg/ml of RA for 4 days). Activation of Shh signalling by RA is known to be conserved among vertebrate limbs/fins (*Riddle et al., 1993*; *Hoffman et al., 2002*; *Dahn et al., 2007*), and as expected, the most reliable Shh target gene, *Ptch1* expression (*Marigo et al., 1996*; *Vokes et al., 2008*; and see *Figure 1—figure supplement 1F* for phylogenetic analysis) was increased and expanded anteriorly (*Figure 4A*). Consistent with this, *Hand2* expression also extended anteriorly (*Figure 4B*)—probably due to inhibition of Gli3 repressor formation by ectopic activation of

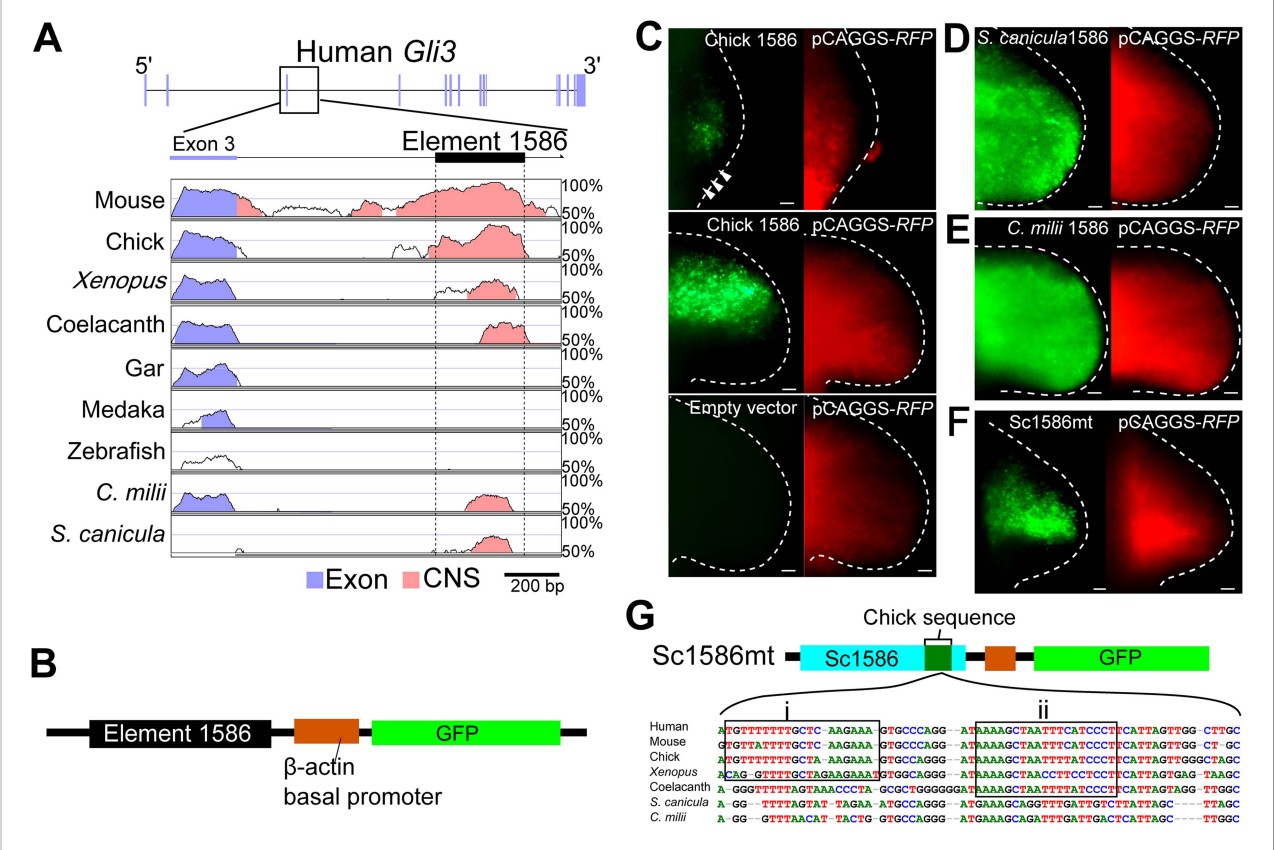

**Figure 3**. The *Gli3* limb-specific enhancer of *S. canicula* and *C. milii*. (**A**) VISTA plots of *Gli3* intron 3 from indicated animals. Blue vertical bars, exons of human *Gli3*; black rectangle, element 1586. Regions with >70% identity are indicated: blue, exon; pink, non-coding sequences. (**B**) The enhancer construct. (**C**) GFP expression in chick forelimb buds driven by chick element 1586 at stage 19 (top, *n* = 3/3), stage 23 (middle, *n* = 14/14), and empty vector (bottom, *n* = 0/7). pCAGGS-*RFP* (right). (**D–F**) GFP expression driven by element 1586 of *S. canicula* (**D**, *n* = 11/11), *C. milii* (**E**, *n* = 10/10) and Sc1586mt (**F**, *n* = 4/4). Scale bars, 100 μm. (**G**) Scheme of Sc1586mt, *S. canicula* enhancer (blue) partially replaced by chick sequence (green) and alignment. Boxes indicate tetrapod (i) and sarcopterygian (ii) specific sequences.

The following figure supplement is available for figure 3:

**Figure supplement 1**. Detailed functional analyses of element 1586.

Shh signalling (as revealed by the extended *Ptch1* expression). On the other hand, *Pax9* expression (an anterior marker) was significantly downregulated and showed only weak expression in the anterior part of the fin buds (*Figure 4C*). The most anterior regions may not be sensitive to this treatment, as expression of *Alx4* was not significantly shifted (*Figure 4D*), and this may be due to the lack of inhibitory regulation from Hand2 to Gli3 described above. To test whether the results of RA treatment were due to specific effects on AP patterning or instead due to a more general interference with limb development, we examined a marker for proximal-distal (PD) patterning in mouse and chick limb buds—*Hoxa13* (*Tamura et al., 1997*; *Mercader et al., 2000*; *Yashiro et al., 2004*). In RA-treated pectoral fin buds, *Hoxa13* expression was weaker than in control, but a shift in its expression domain was not seen (*Figure 4E*), showing that the impact of RA in these experiments is primarily on the AP patterning (the shifts of *Ptch1*, *Hand2*, and *Pax9*, *Figure 4A–C*), rather than on PD patterning or a general impact on development. Most intriguingly, we examined skeletal patterns of *S. canicula* pectoral fins in these partially 'posteriorised' fin buds (*Figure 4F*). Phenotypes varied from mild to severe but in all cases the appearance of distinct anterior elements (propterygium and mesopterygium) was lost. In the mild cases, a proximal element anterior to the metapterygium is attached to the pectoral girdle (single asterisk in *Figure 4F*). This proximal element may result from

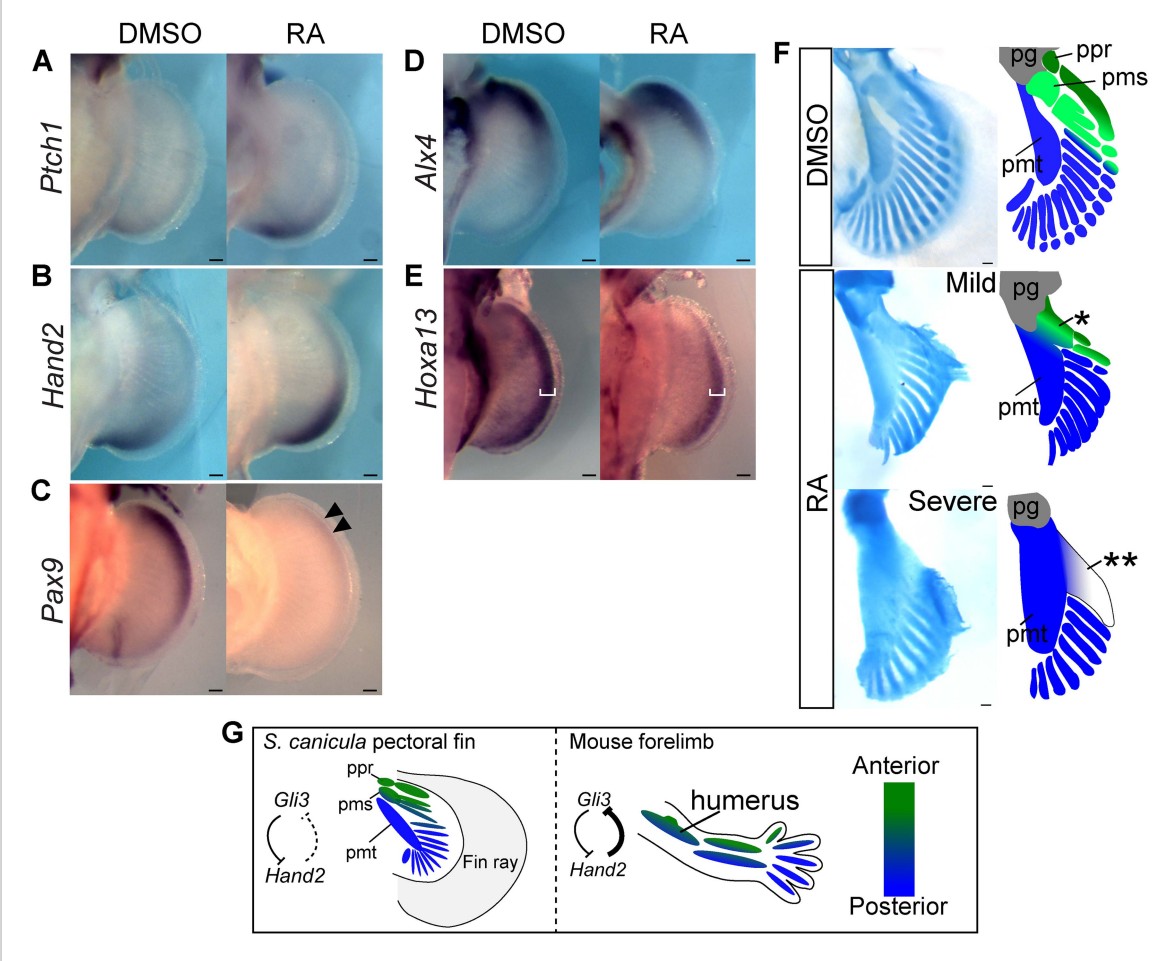

**Figure 4**. RA treatment causes ectopic activation of Shh signalling and loss of anterior skeletal elements. (**A–E**) In situ hybridisation of *S. canicula* pectoral fin buds for *Ptch1* (**A**; left fins flipped horizontally), *Hand2* (**B**), *Pax9* (**C**), *Alx4* (**D**), and *Hoxa13* (**E**) treated with 1% DMSO or 1 μg/ml retinoic acid (RA) (*n* = 2/2 for each for each except *n* = 4/4 for *Hand2*). Arrowheads in **C**, a weak expression of *Pax9*. White brackets in **E**, width of *Hoxa13* expression domain along the proximal-distal (PD) axis. (**F**) Pectoral fin skeletal patterns of 1% DMSO control (*n* = 4) and 1–2 μg/ml RA (*n* = 4). Right panels, schematics of interpretive skeletal patterns. *, an anterior proximal radial; **, a fused radial attached to the metapterygium; pg, pectoral girdle; ppr, pms and pmt, proximal propterygium, mesopterygium and metapterygium. Scale bars, 100 μm. (**G**) Comparison between *S. canicula* fin and mouse limb. Green and blue colours represent anterior–posterior (AP) positional information. ppr, pms, and pmt denote proximal propterygium, mesopterygium, and metapterygium, respectively.

a fusion of the proximal parts of propterygium and mesopterygium. Whereas, the severe cases still have a fused element anterior to the metapterygial axis (double asterisk in *Figure 4F*), but this element is not directly attached to the pectoral girdle, indicating that the pectoral fin of the severe phenotype has lost the anterior proximal elements. By contrast, the posterior metapterygium itself was larger than normal, but retained its strong identity as the primary axis from which radial branching was observed. Although RA treatment potentially cause non-specific effects, given the clear affect on AP patterning (the shifts of *Ptch1*, *Hand2*, and *Pax9*, *Figure 4A–C*), while causing no obvious effects on PD patterning, the main cause of the phenotype is likely caused by the AP pattern change.

In the present study, we have found that *S. canicula* pectoral fin buds have a gross posterior shift in the AP coordinate system compared to mouse limb buds. We show that *S. canicula* and *C. milii* lack a specific enhancer activity for *Gli3*, which in tetrapods mediates the posterior repression, and that this genetic difference likely contributes to the shift of AP positional information. Finally, RA treatment analyses suggest that a partial posteriorisation of *S. canicula* fin buds leads to a loss of anterior proximal elements (propterygium and mesopterygium). Thus, while the loss of the anterior proximal

elements during evolution was associated with cis-regulatory changes of Gli3 in the RA experiments, it was driven by a Shh-mediated affect on the Gli3 protein itself, but in both cases achieving similar phenotypic changes by anterior shift in AP pattern. In support of our observations, a recent study also showed that anterior extension of Shh signalling accompanied with an anterior shift of *Gli3* expression resulted in a loss of anterior skeletal elements in mouse limbs (*Li et al., 2014*). Considering all these data together, we therefore propose that one of the key events during the fin-to-limb transition was an anterior shift of AP positional information (a posteriorisation), which caused the loss of anterior proximal elements (*Figure 4G*).

In the RA treatment experiments, we also observed that the anterior distal radials reduced and only metapterygial radials retained, suggesting that anterior shift in AP positional information may also have had an impact on the distal radials during the fin-to-limb transformation. Interestingly, nearly 30 years ago, a classic study proposed that the distal end of the metapterygial axis (which has a uniformly posterior position in chondrychthians) bent anteriorly during acquisition of digits—the so-called digital arch model (*Shubin and Alberch, 1986*; *Oster et al., 1988*). Although the detailed validity of this model is unclear (*Wagner and Larsson, 2007*), there is a possibility that an AP shift in molecular patterning was involved in the acquisition of digits. In addition to our RA treatment analysis, knockdown analyses of actinotrichia proteins, which are components of fin rays and lost in tetrapod, show an anterior shift in several gene expressions in zebrafish pectoral fin buds (*Zhang et al., 2010*). Therefore, it is interesting to speculate that AP positional information may have shifted several times until the acquisition of digits.

We have shown that the *Gli3* regulatory region of *S. canicula* and *C. milii* lacks the tetrapod-specific repressive element, which is likely needed for the Gli3–Hand2 interaction in mouse limb buds. In mice, *Gli3*$^{-/-}$; *Hand2*$^{-/c}$ limbs show a severe dysplastic humerus (some of them have ectopic protrusion in humerus; *Osterwalder et al., 2014*), suggesting that the Gli3–Hand2 interaction has an important role for patterning the proximal elements. However, how *Gli3* regulates the proximal skeletal pattern is not well understood even in mice. Although *Gli3* is involved in the stylopod (humerus/femur) formation in mice, the phenotype in stylopod always appears with combination of other gene knockouts. For example, *Gli3*$^{-/-}$; *Plzf*$^{-/-}$ mice lack a femur, and *Gli3*$^{-/-}$; *Alx4*$^{-/-}$ mice exhibit humerus malformation (*Barna et al., 2005*; *Panman et al., 2005*). These facts suggest that evolutionary modification of *Gli3* regulation is likely necessary, but additional regulatory modifications are required for the loss of the anterior elements. Since *Alx4* and *Hand2* are expressed in *S. canicula* pectoral fin bud, and *Plzf* is involved only in hindlimb development, currently there is no obvious candidate that would be involved in the loss of propterygium and mesopterygium. Although *S. cacnicula* genome has not been sequenced, systematic studies at genome-wide level such as ChIP-seq in *S. cacnicula* pectoral fin buds would be invaluable to provide a more complete picture of evolutionary mechanism of the loss of the anterior elements in the future.

In conclusion, by taking advantage of the slow evolutionary rates of chondrichthyian genomes, we were able to precisely compare the gene expression, function and regulation between pectoral fin and limb development, and discover a key difference between them. In particular, our study suggest that changes in morphogen interpretation by gene regulatory network mutations may have a major impact on morphological evolution.

## Materials and methods

### Animals

Experiments were performed in accordance with guidelines for animal experiments of Tokyo Tech and CRG, and experiments involved in mice were approved by animal ethics committees of CRG (JMC-07-1001P3-JS). Catshark (*S. canicula*) eggs were incubated at 12–16°C in seawater and staged according to (*Ballard et al., 1993*). *C. milii* eggs and embryos were collected as described (*Takagi et al., 2012*) and staged according to (*Didier et al., 1998*). C52BL/6 (Charles River Laboratories, Wilmington, MA) mouse timed-pregnant females were sacrificed at different days after gestation E11.5. Chicken (*Gallus gallus*) eggs were incubated at 38°C in a humidified incubator until the desired Hamburger–Hamilton (HH) stage (*Hamburger and Hamilton, 1951*) was reached. For in situ hybridisation, embryos were fixed overnight in 4% paraformaldehyde in phosphate-buffered saline, dehydrated in a graded methanol series, and stored in 100% methanol at −20°C.

## Gene isolation and phylogenetic analysis

Total RNA was extracted from stage 24 to 29 *S. canicula* embryos, stage 28 chick embryos and E11.5 mouse embryos using an RNeasy kit (Qiagen, Netherlands). cDNA was synthesised by reverse transcription and used as a template for PCR. Extraction of total RNA and cDNA synthesis from *C. milii* embryos were carried out as described (*Takagi et al., 2012*). To clone *S. canicula* and *C. milii* genes, we used primers that were based on the nucleotide sequences of putative *C. milii* orthologues found in the Elephant Shark Genome Project database (http://esharkgenome.imcb.a-star.edu.sg/) (*Venkatesh et al., 2007*) for *Pax9*, *Alx4*, and *Gli3*; SkateBase (http://skatebase.org/) (*Wang et al., 2012*) for *Hand1*, *Zic3*, *Tbx2*, and *Ptch1*; and GenBank for *Gli2* (EU196410) and *Sox9* (EU241880): *S. canicula Alx4*, 5′-AGGAATGAACGGCGAGACTTG-3′ and 5′-TCATGTTGCCCAAGATATAGC-3′; *S. canicula Pax9*, 5′-GCTGTGTCAGCAAGATACTGG-3′ and 5′-CCGCACTGTATGTCATGTAGG-3′; *S. canicula Gli3*, 5′-CAGCCCAGCAGAATACTACC-3′ and 5′-GAGATCTCAGCGCCATTGATG-3′; *S. canicula Gli2*, 5′-GTAAAGCTTACTCACGACTCG-3′ and 5′-CGTAAGAGTCAGCCGAGCTGATG-3′; *S. canicula Sox9*, 5′-CCCAGGTGCTGAAGGGATAC-3′ and 5′-GGCAGGTACTGGTCGAACTC-3′; *S. canicula Hand1*, 5′-GAGAGCATCAACAGCGCATTCGC-3′ and 5′-TTCCTGGTCCTCAACCTGGTC AG-3′; *S. canicula Zic3*, 5′-GTGGCCATGGCGATGTTACTGGATGGTG-3′ and 5′-GTTTCTCGCCGGTG TGCACTCGGATGTG-3′; *S. canicula Tbx2*, 5′-GACACAGAAACCAGCTTCAGTCACAGTC-3′ and 5′-GAAAGTCGCGATACCCAATGTGGATCAG-3′; *S. canicula Ptch1*, 5′-GAGGTTTCACCTCTCGAT GGGAGAACC-3′ and 5′-CCATACTAATGTGTTCTGTTCCCACTG-3′; *C. milii Gli3*, 5′-GAGATCTC AGCGCCATTGATG-3′ and 5′-GAGATCTCAGCGCCATTGATG-3′. To clone chick and mouse genes, we used primers that were based on the nucleotide sequences of *Pax9* (NM_204912) (*Muller et al., 1996*), *Hoxa13* (NM_204139), and *Tbx2* (NM_009324) (*Bollag et al., 1994a*): chick *Pax9*, 5′-TGAGCGA CACCTCGTCGTACC-3′ and 5′-GGTTATGCGATCCACTGCTA-3′; chick *Hoxa13*, 5′-GTCATGTTCCTC TACGACAAC-3′ and 5′-GGTGGACTTCCAGAGGTGAGG-3′; mouse *Tbx2*, 5′-ATCCTGAACTCCAT GCACAAGTACC-3′ and 5′-GAACTGCTGCCCATGCAGGTGGCTG-3′. The gene fragments were cloned into pBluescript SK− or pCR4 (Invitrogen, Thermo Fisher Scientific Inc., Waltham, MA). The partial coding sequences for *Alx4* (1112 bp), *Pax9* (729 bp), *Gli3* (1937 bp), *Hand1* (465 bp), *Tbx2* (926 bp), *Zic3* (919 bp) and *Ptch1* (791 bp) of *S. canicula* and *Gli3* (428 bp) of *C. milii* have been submitted to GenBank under accession numbers KC507187–9, KF748129, and KP055651-KP055653, KF297620, respectively. Phylogenetic analysis was used to confirm the orthology of newly identified *S. canicula* and *C. milii* genes. Amino acid sequences were aligned using ClustalX (*Thompson et al., 1997*). Regions that could not be aligned were excluded from the analysis. Neighbour-joining phylogenetic trees of amino acid sequence data sets were constructed with MEGA5 (*Tamura et al., 2011*). Bootstrapping was carried out with 1000 replicates.

## Probe synthesis and in situ hybridisation

Chick *Alx4* (NM_204162) was kindly provided by Dr Toshihiko Ogura. Riboprobes for *Hand2* (AY057890) and *Hoxa13* (EU005550) of *S. canicula* and for chick *Alx4* were synthesised as described (*Takahashi et al., 1998*; *Tanaka et al., 2002a*; *Sakamoto et al., 2009a*). The cloned genes described above were used as templates for riboprobe synthesis. Whole-mount in situ hybridisation was carried out as described (*Tanaka et al., 2002a*). *Sox9* expressions were scanned with Optical Projection Tomography (OPT) as described (*Sharpe et al., 2002*) and analysed with Volviewer (*Lee et al., 2006*).

## Gli processing analysis

Human *Gli3* (clone name: pFN21AE1055) and *Gli2* (*Roessler et al., 2005*) were obtained from the Kazusa DNA Research Institute (*Nagase et al., 2008*) and Addgene, respectively. pCAGGS was kindly provided by Dr Toshihiko Ogura and originated from Dr Jun-ichi Miyazaki (*Niwa et al., 1991*). For Western blotting analysis, the N-terminal HaloTag in the human Gli3 construct was replaced with a 6×Myc tag (Myc-hsGli3). Then, the human Gli3 PDD (amino acids 644–842) (*Pan and Wang, 2007*) was replaced with the homologous domain from human Gli2 and *S. canicula* Gli2 and Gli3 by a combination of PCR (*Wurch et al., 1998*) and restriction enzyme digestions. The HEK293 cell line was kindly provided by Dr Masayuki Komada. HEK293 cells were grown in Dulbecco's modified Eagle medium (Sigma–Aldrich, St. Louis, MO) supplemented with 10% foetal bovine serum (Gibco, Thermo Fisher Scientific Inc., Waltham, MA) and penicillin/streptomycin (Sigma–Aldrich) at 37˚C. For Western blot analysis, cells were plated in 6-well plates without penicillin/streptomycin and transfected with 4 µg of constructs using polyethylenimine (GE Healthcare, England) for 3 hr. After the transfection, the medium was changed, and cells were cultured for

24 hr and then treated with 50 µM forskolin (FSK; Sigma) in Dimethyl sulfoxide (DMSO) DMSO or with DMSO alone for 24 hr. Whole-cell extracts were prepared by solubilisation in lysis buffer containing 50 mM Tris-HCl, pH 7.5; 150 mM NaCl; 1 mM ethylenediaminetetraacetic acid (EDTA); 1% Triton X-100; 0.1% Sodium Dodecyl Sulfate (SDS)S; 1% sodium deoxycholate; and protease inhibitor cocktail (Roche, Switzerland). Whole-cell lysates were separated by sodium dodecyl sulphate–polyacrylamide gel electrophoresis and analysed by Western blotting and anti-c-Myc (Sigma–Aldrich), anti-rabbit IgG secondary antibody conjugated with horseradish peroxidase (Jackson ImmunoResearch, West Grove, PA), and enhanced chemiluminescence detection (GE Healthcare).

## Enhancer analysis

The limb-specific *Gli3* enhancer was found with VISTA enhancer browser (http://enhancer.lbl.gov/) (*Visel et al., 2007*). The enhancer ID is hs1586, which is located in *Gli3* intron 3 in the human genome (hg19). For alignment, *Gli3* intron 3 sequences from mouse (*Mus musculus*), chick (*G. gallus*), frog (*Xenopus tropicalis*), coelacanth (*Latimeria chalumnae*), gar (*Lepisosteus oculatus*), medaka (*Oryzias latipes*), and zebrafish (*Danio rerio*) from the Ensembl and Pre Ensembl genome browsers (http://www.ensembl.org/, http://pre.ensembl.org/) were collected. The element 1586 homologue from elephant shark (*C. milii*) was retrieved from the genome assembly (http://esharkgenome.imcb.a-star.edu.sg/) (*Venkatesh et al., 2007*) by using human element 1586 sequence as the query. The GenBank accession number of the *C. milii* element 1586 is AAVX01295166. The *S. canicula* counterpart of element 1586 was amplified by PCR with primers designed from conserved sequences of the upstream exon and the distal part of element 1586: 5′-AGTGGACCCCCGAAATGGCTACATGGACC-3′ and 5′-GAACATCTTCTAATTTACTGGAATCCCAG-3. The amplified fragment was then cloned into pBluescript SK−. The sequence of *S. canicula* element 1586 was deposited in GenBank under accession number KF297619. The alignment was carried out with the SLAGAN method, and overall sequence similarities in the alignment were visualised with mVISTA (*Mayor et al., 2000*; *Brudno et al., 2003*; *Frazer et al., 2004*).

For functional analysis, the element 1586 homologues were isolated from chick and *C. milii* genomes by PCR. The following forward and reverse primers were used: chick element 1586, 5′-CGAGCTCCCTCCTCAGTCATTCAGTTCTGC-3′ and 5′-TGTGTGAGACATACTTTGATC-3′; *C. milii* element 1586, 5′-GAGCTCGTACAGTGATGACTGAAATGGTG-3′ and 5′-GAGATTTCGAGTCTCTTT GATC-3′. The amplified DNA fragments were cloned into pBluescript SK−. To subclone the *S. canicula* 1586 fragment, we used the following primers: 5′-CCGCTCTAGAACTAGCATCAATATGATTTGCT GAG-3′ and 5′-CGGGGGATCCACTAG GCTTCACGAGCATCAGGAAC-3′. The element 1586 sequence from each species was subcloned in front of a chicken β-actin basal promoter that is followed by a *GFP* reporter (*Ochi et al., 2012*). Recombined enhancers were created by PCR. In ovo, electroporation was carried out as described (*Suzuki and Ogura, 2008*). A DNA solution was prepared with Maxi Prep (Qiagen). pCAGGS-*RFP* was kindly provided by Dr Cheryll Tickle. *Gli3* limb enhancers and empty β-actin basal promoter–GFP at ∼6 µg/µl, coloured with ∼3% fast green, and co-electroporated with pCAGGS-*RFP* (∼2 µg/µl) into the presumptive forelimb field of stage 13–14 embryos. A CUY21EDIT II electroporator (BEX Co., Ltd., Japan) was used. Electric pulses consisted of one short pulse (25 V, 0.05 ms) and a 0.1-ms interval, followed by five long pulses (8 V, 10 ms) with 1-ms intervals. The electric pulses were applied during injection of the DNA solution.

## RA treatment

*S. canicula* embryos were removed from their egg shells, then placed into 6-well plates. 4–6 ml of artificial seawater containing penicillin/streptomycin was used for culturing embryos. RA was dissolved in DMSO to 2 mg/ml as a stock solution and diluted in the artificial seawater to 1–2 µg/ml. 1% DMSO in the artificial seawater was used as negative controls. Embryos at stage 28–29 were cultured with RA for 4 days for gene expression analyses. For alcian blue staining, embryos at stage 28–29 were cultured with 1–2 µg/ml RA for 20 days and additional 10–18 days after removing RA. Note that effect of RA is highly dependent on individual embryos. Some batches of embryos were lethal at 2 µg/ml of RA, probably due to season or parents' condition. In this case, embryos were treated with 1 µg/ml of RA.

## Acknowledgements

We thank T Ogura, J Miyazaki, A Kuroiwa, H Ogino, and H Ochi for providing the plasmids; A Tweedale and Station Biologique de Roscoff for collecting *S. canicula* embryos; JA Donald, T Toop, and JD Bell for collecting *C. milii* embryos; M Komada for providing cell lines; M Davy, X Xu, S Kuratani, and C

Tickle for comments; and K Munakata, S Ueda, and N Suda for technical advice. Sequences for *Alx4*, *Pax9*, *Gli3*, *Hand1*, *Tbx2*, *Zic3*, *Ptch1*, and element 1586 from *S. canicula* and *Gli3* from *C. milii* are deposited in GenBank under accession numbers KC507187–9, KF748129, KP055651-KP055653, KF297619, and KF297620. Correspondence and requests for materials should be addressed to M Tanaka (mitanaka@bio. titech.ac.jp) and J Sharpe (james.sharpe@crg.eu). This work was supported in part by the Global COE Program 'Evolving Education and Research Center for Spatio-Temporal Biological Network' from the Ministry of Education, Culture, Sports, Science and Technology (MEXT) to KO and MT, the Japan–Australia Research Cooperative Program to SH, the Grant-in-Aid for Scientific Research on Innovative Areas, the Grant-in-Aid for Scientific Research (B) and the Inamori Foundation to MT, ICREA and the Spanish Ministry of Economy and Competitiveness, 'Centro de Excelencia Severo Ochoa 2013-2017', SEV-2012-0208 to JS and the Centre for Genomic Regulation to KO and JS.

## Additional information

### Funding

| Funder | Grant reference | Author |
|---|---|---|
| Japan Society for the Promotion of Science (JSPS) | Grant-in-Aid for Scientific Research (B) | Mikiko Tanaka |
| Ministry of Education, Culture, Sports, Science, and Technology (MEXT) | Grant-in-Aid for Scientific Research on Innovative Areas | Mikiko Tanaka |
| Japan Society for the Promotion of Science (JSPS) | Japan-Australia Research Cooperative Program | Susumu Hyodo |
| Spanish Ministry of Economy and Competitiveness (MINECO) | Plan Nacional Grant (BFU2010-16428) | James Sharpe |

The funders had no role in study design, data collection and interpretation, or the decision to submit the work for publication.

### Author contributions

KO, Conception and design, Acquisition of data, Analysis and interpretation of data, Drafting or revising the article, Contributed unpublished essential data or reagents; SK, Acquisition of data, Analysis and interpretation of data, Contributed unpublished essential data or reagents; WT, SH, Acquisition of data, Contributed unpublished essential data or reagents; JS, MT, Conception and design, Analysis and interpretation of data, Drafting or revising the article

### Ethics

Animal experimentation: Experiments were performed in accordance with guidelines for animal experiments of Tokyo Tech and CRG, and experiments involved in mice were approved by Animal Ethics Committees of CRS (No. JMC-07-1001P3-JS).

## Additional files

### Major datasets

The following datasets were generated:

| Author(s) | Year | Dataset title | Dataset ID and/or URL | Database, license, and accessibility information |
|---|---|---|---|---|
| Onimaru K, Tanaka M | 2014 | Scyliorhinus canicula Alx4 mRNA, complete cds | http://www.ncbi.nlm.nih.gov/nuccore/KC507187 | Publicly available at the NCBI Nucleotide (Accession no: KC507187). |
| Onimaru K, Tanaka M | 2014 | Scyliorhinus canicula Pax9 mRNA, partial cds | http://www.ncbi.nlm.nih.gov/nuccore/KC507188 | Publicly available at the NCBI Nucleotide (Accession no: KC507188). |

| Author(s) | Year | Dataset title | Dataset ID and/or URL | Database, license, and accessibility information |
|---|---|---|---|---|
| Onimaru K, Tanaka M | 2014 | Scyliorhinus canicula Gli3 mRNA, partial cds | http://www.ncbi.nlm.nih.gov/nuccore/KC507189 | Publicly available at the NCBI Nucleotide (Accession no: KC507189). |
| Onimaru K, Kuraku S, Takagi W, Hyodo S, Tanaka M | 2014 | Scyliorhinus canicula element 1586, genomic sequence, a fin/limb enhancer of Gli3 | http://www.ncbi.nlm.nih.gov/nuccore/KF297619 | Publicly available at the NCBI Nucleotide (Accession no: KF297619). |
| Onimaru K, Kuraku S, Takagi W, Hyodo S, Tanaka M | 2014 | Scyliorhinus canicula Hand1 mRNA, partial cds | http://www.ncbi.nlm.nih.gov/nuccore/KF748129 | Publicly available at the NCBI Nucleotide (Accession no: KF748129). |
| Onimaru K, Kuraku S, Takagi W, Hyodo S, Sharpe J, Tanaka M | 2015 | S. canicula Zic3 mRNA, partial cds | http://www.ncbi.nlm.nih.gov/nuccore/KP055652 | Publicly available at the NCBI Nucleotide (Accession no: KP055652). |
| Onimaru K, Kuraku S, Takagi W, Hyodo S, Sharpe J, Tanaka M | 2015 | S. canicula Tbx2 mRNA, partial cds | http://www.ncbi.nlm.nih.gov/nuccore/KP055651 | Publicly available at the NCBI Nucleotide (Accession no: KP055651). |
| Onimaru K, Kuraku S, Takagi W, Hyodo S, Sharpe J, Tanaka M | 2015 | S. canicula Ptch1 mRNA, partial cds | http://www.ncbi.nlm.nih.gov/nuccore/KP055653 | Publicly available at the NCBI Nucleotide (Accession no: KP055653). |
| Onimaru K, Kuraku S, Takagi W, Hyodo S, Tanaka M | 2015 | C. milii Gli3 mRNA, partial cds | http://www.ncbi.nlm.nih.gov/nuccore/KF297620 | Publicly available at the NCBI Nucleotide (Accession no: KF297620). |

The following previously published datasets were used:

| Author(s) | Year | Dataset title | Dataset ID and/or URL | Database, license, and accessibility information |
|---|---|---|---|---|
| Tanaka M, Munsterberg A, Anderson WG, Prescott AR, Hazon N, Tickle C | 2002 | Scyliorhinus canicula dHand protein mRNA, partial cds | http://www.ncbi.nlm.nih.gov/nuccore/AY057890 | Publicly available at the NCBI Nucleotide (Accession no: AY057890). |
| Sakamoto K, Onimaru K, Munakata K, Suda N, Tamura M, Ochi H, Tanaka M | 2009 | Scyliorhinus canicula HoxA13 mRNA, partial cds | http://www.ncbi.nlm.nih.gov/nuccore/EU005550 | Publicly available at the NCBI Nucleotide (Accession no: EU005550). |
| Muller TS, Ebensperger C, Neubuser A, Koseki H, Balling R, Christ B, Wilting J | 2015 | Gallus gallus paired box 9 (PAX9), mRNA | http://www.ncbi.nlm.nih.gov/nuccore/NM_204912 | Publicly available at the NCBI Nucleotide (Accession no: NM_204912). |
| Takahashi M, Tamura K, Buscher D, Masuya H, Yonei-Tamura S, Matsumoto K, Naitoh-Matsuo M, Takeuchi J, Ogura K, Shiroishi T, Ogura T, Izpisua Belmonte JC | 2013 | Gallus gallus ALX homeobox 4 (ALX4), mRNA | http://www.ncbi.nlm.nih.gov/nuccore/NM_204162 | Publicly available at the NCBI Nucleotide (Accession no: NM_204162). |
| Bollag RJ, Siegfried Z, Cebra-Thomas J, Garvey N, Davison EM, Silver LM | 1994 | Mus musculus T-box 2 (Tbx2), mRNA | http://www.ncbi.nlm.nih.gov/nuccore/120407038 | Publicly available at the NCBI Nucleotide (Accession no: NM_009324). |
| Zhang G, Cohn MJ | 2009 | Scyliorhinus canicula Sox9 (Sox9) mRNA, complete cds | http://www.ncbi.nlm.nih.gov/nuccore/EU241880 | Publicly available at the NCBI Nucleotide (Accession no: EU241880). |

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
