## [Decision Letter]

Thank you for sending your work entitled “A shift in anterior-posterior positional information underlies the fin-to-limb evolution” for consideration at *eLife*. Your article has been evaluated by Diethard Tautz (Senior editor), Marianne Bronner (Reviewing editor), and three reviewers.

The Reviewing editor and the other reviewers discussed their comments before we reached this decision, and the Reviewing editor has assembled the following comments to help you prepare a revised submission.

In this paper, the authors analyze fin development in Chondricthian fish from the catshark, *Scyliorhinus canicula*. Specifically they show a lack of an elaboration of genes in the catfish that are known to impart polarity along the anterior posterior axis of mice and chicken. As this polarity has been tied to the specification and patterning of the tetrapod limb, the authors ask if this shift is genetically regulated within the catshark through *Gli3* transcriptional regulation and further if perturbation of general AP signaling is sufficient for altering chondricthian fin development in a manner consistent with AP polarity of early tetrapod lineages. Of note they discover a functional enhancer for *Gli3* that is retained in tetrapods but not seen in chondricthians that they argue has been a key development in the evolution of limbs.

This is an issue that has received intense interest from both paleontologists and developmental biologists and thus should be of general interest to the readership of *eLife*. Although the paper is interesting and elegantly done, there are several major issues that cloud their conclusions and therefore need to be addressed.

1) The idea that there is a broadening of the AP axis in the evolution of tetrapod limbs is not new. [46] and [35] outlined a metapterygial arch hypothesis that outlined a shift in AP axis as a key mechanism leading to evolution of the tertrapod limb. They supported this hypothesis with developmental and comparative analysis. Thus, finding molecular correlates of this pattern is not altogether unexpected, although the data in the present manuscript are the most extensive analysis of the patterning changes in chondricthians. Surprisingly, these papers are neither cited nor discussed, and must be included.

2) The data on “posteriorizing” of the *Scyliorhinus* fin through retinoic acid is not convincing. The authors present quite elegant experiments that uncover a limb-specific *Gli3* regulator domain. However the cartilage staining of the treated fins does not look anything similar to the author's interpretation – and does not phenocopy an early tetrapod such as *Tiktaalik*. Even if the remnant posterior fragment can be argued to be stylopodial, the authors must explain how this occurred as they have shown the lack of a key functional *Gli3* regulatory domain in *Scyliorhinus* that is necessary for the regulation of polarity in mouse and chick – presumably even after RA treatment. Unlike the elaborate elements formed in the miniskate (9), the data here make it seem like cartilage development has been delayed or inhibited. Even though dHand expansion is expanded after RA treatment the changes in domains are small and this may be due to developmental delay or independent effects of RA such as on proliferation (note the fin size is quite different than controls at the same stage), rather changes in polarity.

3) The expression patterns in the fins are quite late in comparison with comparable sized limbs such that these signaling events may pattern more distal structures. Changes in patterning of proximal bones such as the humerus would be expected to occur earlier, such as HH stage 20 in the chick. Could the authors comment on how such late expression changes would be expected to alter early developmental patterning events in the shark?

4) Correlation of positional value shift (expression boundaries) in fin/limb bud to resultant adult morphology isn't very tight. The one example that we're shown of an experimentally RA-dosed cat shark pectoral fin is juxtaposed with the pectoral fin of *Tiktaalic*. What we're not shown is the adult fin morphology of *Callorhinchus*, which is also dominated by the metapterygium (and its various, more distal, radials). *Callorhinchus* pectoral fin skeletons are morphologically like those of *Tiktaalik* and its various finned relatives.

5) ‘Partial posteriorisation’ (the authors' term) is a repeated phenomenon in the broad spectrum of paired fin evolution. It looks like this results in a ‘necessary but insufficient’ item of the fin-to-limb agenda, and likewise it might or might not be part of the history of chimaeroid fins. For example, how does *Callorhinchus* have a metapterygium-dominated pectoral fin, but seems not to have employed/evolved the positional value shift at the heart of this study?

6) The connection between the part of the paper that paper that documents the changes in *Gli3* expression and function are not clearly linked to the final part of the paper that uses RA treatment to shift AP patterning. The authors need to make a clearer statement about exactly how they link together these observations using the experimental manipulations they have done. It seems that the RA treatment is a very indirect way to try and link together the observations on *Gli3* to the actual effects of shifting AP patterning. The link seems very indirect and RA treatment is bound to have many effects that are independent of the changes in *Gli3*. Would the authors expect that the reduction of *Shh* signaling in a tetrapod increase the number of bones in the anterior part of the limb, or better yet what would happen if they expressed *Gli3* in a more posterior region? At the very least, the authors need to make a stronger link here. Without this, the work would not be particularly compelling of broad interest.

[Editors' note: further revisions were requested prior to acceptance, as described below.]

Thank you for resubmitting your work entitled “A shift in anterior-posterior positional information underlies the fin-to-limb evolution” for peer review at *eLife*. Your revised submission has been favorably evaluated by Diethard Tautz (Senior editor), Marianne Bronner (Reviewing editor), and one of the original reviewers.

Summary:

The authors have addressed the majority of the previous concerns, but one comment should be addressed more directly.

1) The reviewers remain unconvinced of the interpretation of the RA work and identity of the resulting elements. As this is a key component of their conclusions that alteration of RA/Shh signaling can cause reduction of the pro/mesopterygium, similar to the fin-limb morphological transition, it is important that this is clear to the reader to be able to interpret their findings. For example the severe RA treated fin shown has a strongly stained element (comparable to the proximal elements of the DMSO treated fin) on the anterior side – not compatible with a reduction of a pro/meso component of the fin. Given the lightly staining of the treated fins compared with the control, it would be helpful if the authors could make a schematic of their interpretation of the resulting pattern and explain why there are patterning/staining differences.

2) The title for Figure 4 should reflect that it is RA mediated signaling that was tested not Shh. Shh levels or activity were not directly tested by gene over-expression or like means.

3) Results and discussion, sixth paragraph, the conclusions that “loss of the anterior proximal elements during evolution appears partially ‘driven’ by cis regulatory changes” has not been directly shown. What is shown is that changes are associated with morphological transitions and may have been a component of the changes leading to the evolution of these forms. They also may have been secondary and not directly involved.

---

## [Author Response]

*1) The idea that there is a broadening of the AP axis in the evolution of tetrapod limbs is not new.*
[46]
*and*
[35]
*outlined a metapterygial arch hypothesis that outlined a shift in AP axis as a key mechanism leading to evolution of the tertrapod limb. They supported this hypothesis with developmental and comparative analysis. Thus, finding molecular correlates of this pattern is not altogether unexpected, although the data in the present manuscript are the most extensive analysis of the patterning changes in chondricthians. Surprisingly, these papers are neither cited nor discussed, and must be included*.

As suggested, we have added discussion about the digital arch model in the seventh paragraph of the Results and discussion section.

*2) The data on “posteriorizing” of the* Scyliorhinus *fin through retinoic acid is not convincing. The authors present quite elegant experiments that uncover a limb-specific* Gli3 *regulator domain. However the cartilage staining of the treated fins does not look anything similar to the author's interpretation – and does not phenocopy an early tetrapod such as* Tiktaalik*.*

We agree that the RA-treated skeletons are not a direct phenocopy of *Tikitaalik* and we have therefore removed that panel from Figure 4, and also replaced the scheme of evolutionary process into a comparison between specific species (*S. canicula* and mice). Nevertheless, our important observation remains the striking correlation between the anterior shift of gene expression patterns, and the fusion of basal bones into a single metapterygium. In the context of the fin-to-limb transition, this correlation is a very important scientific observation.

*Even if the remnant posterior fragment can be argued to be stylopodial, the authors must explain how this occurred as they have shown the lack of a key functional* Gli3 *regulatory domain in* Scyliorhinus *that is necessary for the regulation of polarity in mouse and chick – presumably even after RA treatment.*

We agree that this part may have been confusing – in particular we may not have distinguished clearly enough between regulation at the transcriptional level versus the protein activity level. On the one hand, we show that *Gli3* in *S. canicula* is lacking a relevant cis-regulatory element to restrict its expression anteriorly. On the other hand, we also show that RA is indeed able to cause a shift in AP patterning for *Ptch1*, *Hand2* and *Pax9*. However, there is no contradiction here: The knowledge that RA treatment increases Shh signalling in fin and limb buds is well-known from the literature ([40] Cell 75: 1401-1416; [18] Int. J. Dev. Biol*.* 46: 949-956; [9] Nature 445: 311-314). In turn, the effect of Shh signalling on *Gli3* is not an alteration in transcriptional regulation, but rather a modification of the protein, specifically inhibiting formation of the repressor form (which is responsible for repressing *Hand2* in the anterior tissue; [60] Genes Dev. 22: 2651-2663). This reduced repressive activity allows *Hand2* to be upregulated, causing the general AP shift. Thus we propose that the effect seen in the RA-treated embryos is a post-translational regulation of *Gli3* – not transcriptional. In other words, the AP shift during evolution was partially driven by cis-regulatory changes, while by contrast in our RA experiments it was driven by a Shh-mediated effect on the *Gli3* protein itself, but in both cases achieving similar phenotypic changes. We have made this much clearer in the main manuscript now (Results and discussion, fifth and sixth paragraphs).

*Unlike the elaborate elements formed in the miniskate (*[9]*), the data here make it seem like cartilage development has been delayed or inhibited. Even though dHand expansion is expanded after RA treatment the changes in domains are small and this may be due to developmental delay or independent effects of RA such as on proliferation (note the fin size is quite different than controls at the same stage), rather changes in polarity*.

We agree it is important to address the potential non-specific effect of RA experiments, for example on general growth and development of the bud (even in this non-model system which is non-trivial to work with). We have therefore performed new experimental works to address this issue (Figure 4 and Results and discussion, fifth paragraph).

Firstly, we addressed whether the treated fin buds appear to have retarded patterning, by examining another molecular marker which reveals the progress of proximo-distal patterning – *Hoxa13*. We detected slightly weaker expression in RA treated embryos than that in control embryos. However, the expression domain of *Hoxa13* was not obviously shifted (Figure 4), showing that this dosage and timing of RA treatment is having a clear effect on AP patterning (the shifts of *Ptch1*, *Hand2* and *Pax9,*
Figure 4)*,* while causing no obvious effects on PD patterning or general growth.

Secondly, we re-evaluated the sizes of fin buds from the RA-treated experiments, and also repeated the experiment to increase the replicates of *Hand2* analysis. The image in the original panel of this figure was slightly distorted due to the in situ hybridisation process. From further analysis of all RA-treated results, it is clear that we observed the anterior expansion of *Hand2* expression in buds of a similar size to the controls (in Figure 4).

Since the reviewer considered that the change in *Hand2* expression was small, we have also chosen to boost the result by analysis of another AP marker – *Pax9* which is expressed in the anterior tissue (Figure 4). As with *Hand2* and *Ptch1*, this marker also displays a “posteriorisation” (or anterior shift of positional information) as it’s graded levels are reduced after RA treatment. Also important to note is that again the size of the fin bud is not reduced.

*3) The expression patterns in the fins are quite late in comparison with comparable sized limbs such that these signaling events may pattern more distal structures. Changes in patterning of proximal bones such as the humerus would be expected to occur earlier, such as HH stage 20 in the chick*. *Could the authors comment on how such late expression changes would be expected to alter early developmental patterning events in the shark?*

Indeed, we agree that the timing of these patterning processes is important point to address, and so we have added new gene expression data to clarify the time point at which the proximal skeletal elements begin to be patterned. Figure 1—figure supplement 2 now shows that *Sox9* expression starts in the proximal elements between stage 28 and late stage 29. This is the same period for which we show the more posteriorly-biased gene expression patterns in *S. canicula* (Figure 1–figure supplement A-D) and also the RA-treatment experiments (Figure 4).

*4) Correlation of positional value shift (expression boundaries) in fin/limb bud to resultant adult morphology isn't very tight. The one example that we're shown of an experimentally RA-dosed cat shark pectoral fin is juxtaposed with the pectoral fin of* Tiktaalic*. What we're not shown is the adult fin morphology of* Callorhinchus*, which is also dominated by the metapterygium (and its various, more distal, radials).* Callorhinchus *pectoral fin skeletons are morphologically like those of* Tiktaalik *and its various finned relatives.*

Although the fin skeleton of *Callorhinchus* has a strong metapterygium, it also clearly has a large propterygium, which is in contrast to *Tiktaalik*, which only has a single proximal element (metapterygium/ stylopod). To address this comment we have therefore followed the reviewers’ suggestion, and generated a new cartilage staining of *Callorhinchus* and included it in Figure 2.

*5) ‘Partial posteriorisation’ (the authors' term) is a repeated phenomenon in the broad spectrum of paired fin evolution. It looks like this results in a ‘necessary but insufficient’ item of the fin-to-limb agenda, and likewise it might or might not be part of the history of chimaeroid fins. For example*, *how does* Callorhinchus *have a metapterygium-dominated pectoral fin, but seems not to have employed/evolved the positional value shift at the heart of this study?*

We agree that the loss of anterior elements is a repeated phenomenon. However, we do not believe this has happened in the *Callorhinchus milii* pectoral fin skeleton (which we now show in Figure 2), because its skeleton still has a large proximal propterygium (annotated as “pro”). We therefore see no clear evidence against our hypothesis that an AP shift was involved in the fin-to-limb transition.

*6) The connection between the part of the paper that paper that documents the changes in* Gli3 *expression and function are not clearly linked to the final part of the paper that uses RA treatment to shift AP patterning. The authors need to make a clearer statement about exactly how they link together these observations using the experimental manipulations they have done. It seems that the RA treatment is a very indirect way to try and link together the observations on* Gli3 *to the actual effects of shifting AP patterning. The link seems very indirect and RA treatment is bound to have many effects that are independent of the changes in* Gli3*.*

We agree that RA treatment is an indirect way to perturb AP patterning, although we must point out (a) that experimental embryology in a non-model species like this is challenging, and also (b) that precisely this method for boosting Shh signalling has been used in important previous papers about fin patterning in the mini-skate ([9] Nature 445: 311-314) although here we are drawing novel and distinct conclusions about the shift in AP patterning (which nevertheless do not contradict this previous paper).

To further address this general point, we have now strengthened and clarified the manuscript in the following ways (Results and discussion, fifth and sixth paragraphs):

Firstly, we realised that our proposal may have been confusing (the link between the results on *Gli3* regulation and the RA experiments) because in Figure 3 we show that *S. caniculus* lacks the cis-regulatory elements to be inhibited in the posterior tissue, and yet in the RA experiments we report that an AP shift is induced. This point is discussed above (point 2). Essentially, we did not explain the known relation between *Gli3* and RA, and we did not distinguish strongly enough in the text between transcriptional regulation and post-translational regulation. We have now re-written this to make it clear: At the genetic/evolutionary level, our results suggest that acquisition of element 1586 was a way to shift *Gli3* expression anteriorly. However, in the experiments extensive Shh signalling triggered by RA causes a similar AP shift, but through a different mechanism inhibiting production of *Gli3* repressor form (rather than altered regulation of *Gli3* transcription). The important result is that we have two cases of an anterior shift: (a) the comparison between sharks and tetrapods, (b) the RA-treated shark buds, and in both cases two important observations are correlated: (i) the anterior shifting of genes, (ii) a reduction or loss of anterior basal bones, leaving a metapterygium-dominated arrangement.

Secondly, we have addressed the problem of possible non-specific effects of RA (as described above for point 2). We have examined *Hoxa13* expression in RA treated embryos to assess possible effects on proximo-distal (PD) patterning, which is another major role of RA in tetrapod limb buds. Although expression of *Hoxa13* in RA treated embryos was slightly weaker than in control embryos, nevertheless PD patterning appears hardly affected compared to the clear AP shifts in *Hand2* and *Pax9* expression (as the PD boundary of *Hoxa13* expression is the same as controls, Figure 4).

*Would the authors expect that the reduction of* Shh *signaling in a tetrapod increase the number of bones in the anterior part of the limb, or better yet what would happen if they expressed* Gli3 *in a more posterior region? At the very least, the authors need to make a stronger link here. Without this, the work would not be particularly compelling of broad interest*.

*Gli3* overexpression in a posterior limb bud may not be enough to increase the number of the proximal elements (stylopod), because the phenotype in the stylopod always appears in combination with *Gli3* and other gene knockouts. Nevertheless, it is clear that in these cases the expected result (of element reduction) is indeed seen: *Gli3*^-/-^;*Plzf*^-/-^ mice lack the femur ([4] Nature 436, 277-281), *Gli3*^-/^;*Alx4*^-/-^ mice show reduction of humerus ([38] Int. J. Dev. Biol*.* 49, 443-448). Therefore, to increase the number of stylopod elements (i.e. to get limbs back to the ancestral condition), *Gli3* and additional genes are likely required. Since *Alx4* and *Hand2* are already expressed in the *S. canicula* pectoral fin bud, and *Plzf* is involved only in the hindlimb bud, we do not have any candidates that might be involved in the fin-to-limb evolution. Although *S. canicula* genome is not sequenced yet, systematic studies at whole genome level such as ChIP-seq analysis in *S. canicula* fin bud would provide a more complete picture of evolutionary mechanism of the loss of the anterior elements in the future (Results and discussion, eighth paragraph).

[Editors' note: further revisions were requested prior to acceptance, as described below.]

*1) The reviewers remain unconvinced of the interpretation of the RA work and identity of the resulting elements. As this is a key component of their conclusions that alteration of RA/Shh signaling can cause reduction of the pro/mesopterygium, similar to the fin-limb morphological transition, it is important that this is clear to the reader to be able to interpret their findings. For example the severe RA treated fin shown has a strongly stained element (comparable to the proximal elements of the DMSO treated fin) on the anterior side – not compatible with a reduction of a pro/meso component of the fin. Given the lightly staining of the treated fins compared with the control, it would be helpful if the authors could make a schematic of their interpretation of the resulting pattern and explain why there are patterning/staining differences*.

As suggested, we have added a schematic of our interpretation to Figure 4. In particular, since the element on the anterior side of metapterygium in the severe phenotype is not directly attached to the pectoral girdle, we interpret it as a fused radial attached to the metapterygium (indicated by ** in Figure 4). Therefore, we believe that the skeletal pattern of the severe phenotype represents a loss of anterior proximal elements. We have added the details of our interpretation to the main text as well (Results and discussion, fifth paragraph).

The different levels of staining between controls and RA treated fins seem to reflect individual differences in exposure time to Alcian Blue, as the other control sample show a lighter staining. We have therefore replaced the control figure with the lighter stained sample (Figure 4).

*2) The title for*
Figure 4
*should reflect that it is RA mediated signaling that was tested not Shh. Shh levels or activity were not directly tested by gene over-expression or like means*.

We have changed the title of Figure 4 into “RA treatment causes ectopic activation of Shh signaling and loss of anterior skeletal elements.”

*3) Results and discussion, sixth paragraph, the conclusions that “loss of the anterior proximal elements during evolution appears partially ‘driven’ by cis regulatory changes” has not been directly shown. What is shown is that changes are associated with morphological transitions and may have been a component of the changes leading to the evolution of these forms. They also may have been secondary and not directly involved*.

As suggested, we have changed the conclusion into “while the loss of the anterior proximal elements during evolution was associated with cis-regulatory changes of *Gli3*…” (Results and discussion, sixth paragraph).